# Quality and Diversity Both Matters When Merging Models

## Abstract

Generalization to distribution shifts is a primary goal in modern machine learning literature. Ensemble methods, including both output-space ensemble and weight-space ensemble (model merging), are renowned for their robust generalization capabilities over multi-task settings, leveraging the diverse features from source models to improve cross-task transferability. While most studies on model merging focus on constructing diverse pools of task vectors obtained from foundation models trained on different tasks, we also emphasize the quality of each source. In this paper, we introduce a novel method for selectively merging task vectors to achieve superior generalization on target domains. Our approach uniquely considers both the diversity and quality of individual models. Using Determinantal Point Processes (DPP), we propose a probabilistic framework that optimally selects which models to average in a plug-and-play manner, ensuring a balanced consideration of quality and diversity. Theoretical support is provided for our hypothesis that this dual consideration yields a tighter generalization error bound for the unified model. Empirically, we present experiments in an out-of-distribution setting where there is significant violation in identically distributed conditions between the source and target domains.

## Introduction

In the modern era of machine learning, addressing the challenge of distribution shift is crucial, as the assumption of identical distribution between source and target domains may not hold in real-world scenarios. The importance of this issue is magnified in the context of large-scale, foundation models that are typically fine-tuned on diverse sources of datasets. A promising solution to this challenge is merging deep learning models together. Deep ensembles, which are combinations of diverse models, are known for their ability to generalize well on distribution shifts due to diverse features from the source models, enhancing the model's ability to transfer across various tasks. In practical settings, we already possess a variety of fine-tuned models with bless of foundation models and abundance of datasets. Therefore, selectively averaging weights based on quality and diversity in a training-free manner becomes essential.

In this paper, we introduce a novel method for selectively averaging neural networks to achieve solutions with superior generalization on target domains. Unlike existing methods, our approach explicitly considers both the diversity and quality of individual models. We propose a probabilistic framework that optimally selects models to average, ensuring a balanced consideration of both quality and diversity. We provide theoretical support for the hypothesis that considering both quality and diversity yields a tighter generalization error bound for the averaged model.

The contributions of this research are summarized as follows:

- We introduce a novel model merging strategy to exploit both the quality and diversity of source models.
- We provide a generalization error bound for ensemble classifier, supported by theoretical proof.
- Our method demonstrates superior performance in non-i.i.d. settings where the assumption of identical distribution in the source domain is violated.

## Preliminaries and Related Works

### Averaging Model Weights

Averaging the weights of models is a powerful approach in finding good solution in deep learning. (Izmailov et al. 2018) posits that averaging weights leads to wider optima in the loss surface, thereby enhancing generalization ability. By simply averaging multiple checkpoints during the training process, the solution tends to converge to flatter minima compared to the traditional Stochastic Gradient Descent solution. Diverse Weight Averaging for Out-of-Distribution (OOD) Generalization (DiWA) (Rame et al. 2022) averages weights obtained from independent training runs that share the same initialization, thereby increasing functional diversity across the averaged models. This work explains the success of weight averaging in OOD scenarios by highlighting the empirical similarity between weight averaging and output ensembling.

Due to the abundance of fine-tuned models and efficiency of foundation models, modern weight averaging methodologies utilize diverse fine-tuned models. Model Soup (Wortsman et al. 2022a) averages diverse fine-tuned weights that vary across hyperparameter configurations yielding good generalization ability under distribution shift. Model Ratatouille (Daheim et al. 2023) recycles diverse fine-tuned models for OOD generalization. This approach aims to maximize weight diversity by leveraging the diversity in aux-

iliary tasks. It averages multiple weights fine-tuned from different initializations, each trained on different auxiliary tasks. The rationale for this ensemble's improved generalization in OOD scenarios is that fine-tunings of the same pre-trained foundation model are linearly connected in the loss landscape, despite different initializations, thus allowing successful averaging and yielding a flatter solution.

## Model Merging for Multi-Task Learning (MTL)

Recently, impressed by arithmetic of embedding vectors in language models, weight averaging has been extended to merging task vectors (Ilharco et al. 2022), which are obtained by subtracting pre-trained weight from task-specific fine-tuned weights. Such extension enabled semantic insights on MTL. Several approaches have extended the idea of merging task vectors using various heuristics, such as resolving interference due to redundant parameter values and aligning signs of weights (Yadav et al. 2024), or preserving the important parameters defined via Fisher Information Matrix (Matena and Raffel 2022). (Wang et al. 2024) extended typical 8 computer vision classifications up to 20 tasks, while proposing novel heuristic by eliminating exclusively task-specific weights to improve general performance in MTL. To overcome its limitation in using equal merging coefficients, (Yang et al. 2023) introduced test time adaptation and layer-wise merging. (Tang et al. 2024b) introduced MLP layer of Transformer to flexibly adapt to test tasks and its experiments extensively discussed the generalization and robustness capability of merging models in MTL.

## Determinantal Point Processes

Determinantal Point Processes (DPP) have gained considerable attention in the machine learning community due to their ability to model diversity and provide elegant solutions for subset selection problems. Originally introduced in the context of quantum physics, DPPs have since found applications in a variety of fields including computer vision (Elfeki et al. 2019), information retrieval (Song et al. 2018), and recommendation systems (Liu, Walder, and Xie 2022). One of the pioneering works in applying DPPs to machine learning is done by (Kulesza, Taskar et al. 2012), which provided a comprehensive framework for DPPs and demonstrated their effectiveness in diverse subset selection tasks.

Given a set of data $\mathcal{Y} = x_1, \ldots, x_N$, a point process $\mathcal{P}$ is a probability measure over the set of all subsets of $\mathcal{Y}$. $\mathcal{P}$ is a DPP if a random subset $\mathbf{Y}$ sampled according to $\mathcal{P}$ satisfies:

$$\mathcal{P}_{\mathbf{L}}(\mathbf{Y} = Y) = \frac{\det(\mathbf{L}_Y)}{\sum_{Y \subset \mathcal{Y}} \det(\mathbf{L}_Y)} = \frac{\det(\mathbf{L}_Y)}{\det(\mathbf{L} + \mathbf{I})} \propto \det(\mathbf{L}_Y)$$

The DPP kernel $\mathbf{L}$ is characterized by a similarity matrix $\mathbf{S}$, where $S_{ij}$ defines the similarity between two items $(\mathbf{x}_i, \mathbf{x}_j)$.

# Methodology

## Notation

Let $T$ represent the target or test domain and $S$ represent the source or train domain. The distribution of the source domain is denoted as $D_S = \{(x_i, y_i)\} \sim S$. We define $h \in$

$\mathcal{H}$ as a sampled classifier or hypothesis, where $h : \mathcal{X} \to \mathcal{Y}$ and $h \approx f(x, \theta_m)$. Here, $f$ is the labeling function (ground truth) parametrized by $\theta_m$.

The classification performance of $h$ for a single data point $(x, y)$ is measured by $\ell(h(x), y)$. The expected loss (risk function) over all data points for an arbitrary data distribution $D$ is defined as $\mathcal{L}_D(h) = \mathbb{E}_{(x,y) \sim D}[\ell(h(x), y)]$, assuming that $\mathcal{L}(h)$ is convex with a range of $[0, 1]$.

The parameter or weight specifying each classifier $h$ is denoted by $\theta$, thus $h = h(\cdot; \theta)$. Finally, $\rho$ represents the ensemble distribution, or ensemble strategy.

**Problem Setup** Given $N$ (fine-tuned) models and source data, we aim to choose $M$ models to generate ensemble that can generalize well on target domain $T$.

## Generalization Error Bound

We suggest the generalization error bound of selectively merged neural network classifier as follows:

**Proposition 1.** *Target risk of weight averaged model is bounded by source risk of individual models and diversity of softmax outputs.*

$$\mathcal{L}_T(h_{WA}) = \mathcal{L}_T(h_{ENS}) + O(\Delta^2)$$

$$\leq \frac{1}{M} \sum_{m=1}^{M} \mathcal{L}_{S_m}(\theta_m) - \mathbb{D}(\rho) + d_1(D_S, D_T) + \nu + O(\Delta^2)$$

- $\mathcal{L}_{S_m}(\theta_m)$ *is source risk of $m$-th model.*
- $\mathbb{D}(\rho)$ *is diversity of the ensemble of selected models.*
- $d_1(D_S, D_T)$ *is the divergence between source and target domain.*
- $\nu$ *is the difference in labeling functions across the two domains.*
- $O(\Delta^2)$ *is an approximation error between weight-space averaging and output-space averaging.*

We start by approximating weight average $h_{WA}$ with output ensemble $h_{ENS}$. Various researches on weight averaging including (Izmailov et al. 2018), (Wortsman et al. 2022b), and (Rame et al. 2022) have shown the relationship between weight-space averaging and output-space averaging by employing Taylor expansion.

**Lemma 1.** *Suppose we are given $\{\theta_m\}_{m=1}^{M}$ with $M$ fine-tuned models.*
*Denoting $\Delta_{\theta_M} = \max_m \|\theta_m - \theta_{WA}\|_2$, $\forall(x, y) \in \mathcal{X} \times \mathcal{Y}$:*

$$h_{WA}(x) = h_{ENS}(x) + O(\Delta_{\theta_M}^2),$$
$$\ell(h_{WA}(x), y) = \ell(h_{ENS}(x), y) + O(\Delta_{\theta_M}^2).$$

Then, according to (Ben-David et al. 2010), we bound target risk with respect to individual source risks and the divergence between two distributions. Here, the divergence and the constant $\nu$ is irreducible.

**Lemma 2.** *For a ensemble classifier (hypothesis) $h_{ENS}$,*

$$\mathcal{L}_T(h_{ENS}) \leq \mathcal{L}_S(h_{ENS}) + d_1(D_S, D_T) + \nu,$$

$$\nu = \min \{\mathbb{E}_{D_S}[|f_S(x) - f_T(x)|], \mathbb{E}_{D_T}[|f_S(x) - f_T(x)|]\}$$

Now, adapted from (Ortega, Cabañas, and Masegosa 2022), we decompose the loss of an ensemble classifier to derive diversity out of the function. Prediction ensemble classifier $h_{\text{ENS}}$, is defined using the $\rho$-weighted model average predictor and the cross-entropy loss. In this ensemble, the individual models are probabilistic classifiers whose output is a conditional distribution over the class labels $\mathcal{Y}$ given the sample $x$, i.e. $h(x;\theta) = p(\cdot|x)$. Thus, for an specific input $(x, y)$, the loss of an individual predictor is defined as $\ell_{ce}(x, y; \theta) = -\log p(y|x; \theta)$ and the loss of this ensemble is $\ell_{ce}(x, y; \rho) = -\log \mathbb{E}_{\rho}[p(y|x; \theta)] = -\log \frac{1}{M} \sum_{m=1}^{M} p(y|x; \theta_m)$. Note that in weight averaging scenario, ensemble distribution $\rho$ indicates discrete uniform distribution over M fine-tuned models.

**Lemma 3.** *Under the given setting,*

$$\mathcal{L}_S(h_{ENS}) \leq \mathbb{E}_\rho[\mathcal{L}_{S_m}(\theta_m)] - \mathbb{D}(\rho),$$

*where* $\mathbb{D}(\rho) = \mathbb{E}_D\left[\mathbb{V}_\rho\left(\frac{p(y|x; \theta_m)}{\sqrt{2} \max_{\theta_m} p(y|x; \theta_m)}\right)\right]$

The proposed error bound of the weight-averaged classifier implies small individual source risk and high diversity over ensemble tightens the error bound at the target domain.

## Kernel Construction

The core of our methodology involves constructing the likelihood kernel of the source models. This kernel explicitly measures the quality and diversity of each source model. Given a large pool of fine-tuned models under the non-i.i.d. assumption, DPP select which weights to average proportional to the determinant of the subset. The decomposition of the DPP kernel $\mathbf{L}$ intuitively constructs diverse subsets while weighing each model according to a unary metric $\mathbf{q}$. We define the likelihood kernel $\mathbf{L}$ in terms of the Gram matrix $\mathbf{L} = \mathbf{V}^T\mathbf{V}$, where $\mathbf{L}$ can be decomposed as a quadratic form of the quality vector $\mathbf{q}$ and the similarity kernel $\mathbf{S}$:

$$\mathbf{L} = \text{diag}(\mathbf{q}) \cdot \mathbf{S} \cdot \text{diag}(\mathbf{q})$$

$$\mathbf{L}_{i,j} = q_i \mathbf{S}_{i,j} q_j$$

Here, $q_i$ represents the quality of the model fine-tuned with $i$-th task, and $\mathbf{S}_{i,j}$ is the similarity between $i$-th task and $j$-th task.

**Quality.** We define $q_i$ as the relative increase in the validation accuracy of the fine-tuned model using $i$-th task compared to the pre-trained model:

$$q_i = \frac{\text{Accuracy}_{ft}^i - \text{Accuracy}_{pt}}{\text{Accuracy}_{pt}}$$

$\mathbf{q}$ is normalized to unary metric so that $||\mathbf{q}|| = 1$.
**Similarity.** We define $\mathbf{S}_{i,j}$ to be the average cosine similarity between representation of entire validation samples obtained from models fine-tuned with $i$-th task and $j$-th task. The quality term and similarity metric are inspired from the generalization error bound discussed in Proposition 1.

## Experiment

### Experimental Setup

**Models and Datasets** We use the pre-trained CLIP-ViT-B/32 model (Radford et al. 2021), which has been trained on a large-scale dataset consisting of image-text pairs. This model is capable of performing open-vocabulary image classification. As source datasets, we employed SUN397 (Xiao et al. 2010), Stanford Cars (Krause et al. 2013), RESISC45 (Cheng, Han, and Lu 2017), DTD (Cimpoi et al. 2014), SVHN (Netzer et al. 2011), GTSRB (Stallkamp et al. 2012), MNIST (LeCun et al. 1998), and EuroSAT (Helber et al. 2019) to produce fine-tuned models.
**Baselines** We compare our approach with the model merging techniques proposed in Task Arithmetic (Ilharco et al. 2022), and Ties-Merging (Yadav et al. 2024), who did not adopt test time adaptation setup and chose best scaling coefficient over 0.0~1.0. The experiment was conducted using Fusion Bench (Tang et al. 2024a) framework.

### Results

In this section, we demonstrate that selectively merging models based on their quality and diversity enhances generalization and robustness capabilities. The generalization experiment evaluates the model's ability to perform well on unseen domains, while the robustness experiment assesses the model's performance under distributional shifts. Test distribution shifts were introduced following the strategy of (Hendrycks and Dietterich 2019), incorporating seven different corruptions along with the original (clean) data.
**Generalization experiments.** To evaluate whether quality- and diversity-aware selection improves generalization, we applied the Determinantal Point Process (DPP) heuristic to select models to merge across six seen tasks. The selected task vectors were merged using baseline methods in a plug-and-play manner. For Task Arithmetic, DPP selected five tasks, excluding SUN397. For TIES-Merging, it selected five tasks, excluding Stanford Cars. As shown in Table 1, selectively merging task vectors improved average accuracy on unseen tasks (EuroSAT and MNIST) for both baselines. Additionally, the average accuracy across all eight tasks surpassed that of baseline methods.
**Robustness experiments.** To examine the robustness of the proposed merging strategy, we applied it to datasets with seven corrupted test conditions to introduce domain shifts. In this setup, TIES-MERGING (4) denotes merging fine-tuned task vectors corresponding to four test tasks, while TIES-MERGING (8) indicates merging vectors from all eight tasks. Our method, using DPP, selected six task vectors, excluding RESISC45 and Stanford Cars. As shown in Table 2, selective merging outperformed both merging all task vectors and merging only in-domain task vectors consistently in all corrupted setups. This demonstrates that leveraging selective sources based on quality and diversity provides a superior strategy, effectively tightening the generalization error bound.
Additionally, we investigated whether DPP effectively samples diverse subsets of task vectors for merging. Figure 1

| METHOD | SEEN TASKS | | | | | | | UNSEEN TASKS | | | AVG. |
|---|---|---|---|---|---|---|---|---|---|---|---|
| | SUN397 | CARS | RESISC45 | DTD | SVHN | GTSRB | AVG. | MNIST | EUROSAT | AVG | |
| TASK ARITHMETIC | 0.6338 | 0.6227 | 0.7532 | 0.5798 | 0.8466 | 0.804 | 0.7067 | 0.7727 | 0.4559 | 0.6143 | 0.6836 |
| **TASK ARITHMETIC+DPP** | 0.568 | 0.6478 | 0.7775 | 0.5936 | 0.8534 | 0.8238 | **0.7107** | 0.773 | 0.4756 | **0.6243** | **0.6891** |
| TIES-MERGING | 0.6719 | 0.6573 | 0.7737 | 0.5718 | 0.8852 | 0.8445 | **0.7341** | 0.7929 | 0.3670 | 0.5800 | 0.6955 |
| **TIES-MERGING+DPP** | 0.6853 | 0.5504 | 0.7938 | 0.5835 | 0.8922 | 0.8568 | 0.727 | 0.7916 | 0.4219 | **0.6067** | **0.6969** |

Table 1: Generalization results on two unseen tasks when merging ViT-B/32 models on six tasks.

| METHOD | CARS | EUROSAT | RESISC45 | GTSRB | AVG. | CARS | EUROSAT | RESISC45 | GTSRB | AVG. |
|---|---|---|---|---|---|---|---|---|---|---|
| | CLEAN TEST SET | | | | | CORRUPTED TEST SET (MOTION BLUR) | | | | |
| TIES-MERGING (4) | 65.2 | 83.3 | 78.1 | 67.4 | 73.5 | 64.4 | 53.9 | 76.4 | 57.1 | 62.9 |
| TIES-MERGING (8) | 63.1 | 73.1 | 73.4 | 76.2 | 71.5 | 61.1 | 47.5 | 70.3 | 65.3 | 61.05 |
| **TIES-MERGING+DPP** | 64.5 | 77.0 | 77.4 | 83.1 | **75.5** | 62.3 | 52.0 | 74.4 | 73.6 | **65.6** |
| | CORRUPTED TEST SET (IMPULSE NOISE) | | | | | CORRUPTED TEST SET (GAUSSIAN NOISE) | | | | |
| TIES-MERGING (4) | 60.2 | 45.6 | 69.8 | 38.3 | 53.5 | 61.8 | 47.3 | 73.1 | 42.3 | 56.1 |
| TIES-MERGING (8) | 59.5 | 46.9 | 65.5 | 52.6 | 56.1 | 61.0 | 43.6 | 68.0 | 56.1 | 57.17 |
| **TIES-MERGING+DPP** | 59.9 | 51.2 | 70.1 | 59.9 | **60.3** | 61.51 | 48.81 | 72.43 | 63.42 | **61.54** |
| | CORRUPTED TEST SET (PIXELATE) | | | | | CORRUPTED TEST SET (SPATTER) | | | | |
| TIES-MERGING (4) | 3.3 | 31.8 | 18.0 | 58.5 | 27.9 | 61.3 | 52.9 | 70.3 | 48.1 | 58.2 |
| TIES-MERGING (8) | 2.9 | 31.7 | 15.0 | 72.9 | 30.62 | 59.6 | 52.5 | 66.2 | 65.8 | 61.03 |
| **TIES-MERGING+DPP** | 2.9 | 34.4 | 15.4 | 80.0 | **33.2** | 58.6 | 56.4 | 68.6 | 76.9 | **65.1** |
| | CORRUPTED TEST SET (CONTRAST) | | | | | CORRUPTED TEST SET (JPEG COMPRESSION) | | | | |
| TIES-MERGING (4) | 64.2 | 52.4 | 74.8 | 63.5 | 63.7 | 65.0 | 59.5 | 77.9 | 53.2 | 63.9 |
| TIES-MERGING (8) | 60.9 | 47.3 | 68.4 | 70.9 | 61.88 | 62.9 | 54.1 | 73.2 | 67.0 | 64.3 |
| **TIES-MERGING+DPP** | 62.4 | 52.0 | 72.9 | 78.9 | **66.6** | 64.5 | 58.3 | 77.3 | 73.5 | **68.4** |

Table 2: Robustness results on four corrupted tasks when merging ViT-B/32 models.

presents two-dimensional embeddings of eight datasets obtained from models fine-tuned on Figure 1(a) GTSRB and Figure 1(b) RESISC45. A notable trend is the similarity in embeddings between RESISC45 (brown) and SUN397 (pink), which largely overlap. The selection of SUN397 but not RESISC45 by DPP indicates that the heuristic effectively samples diverse subsets of tasks for merging.

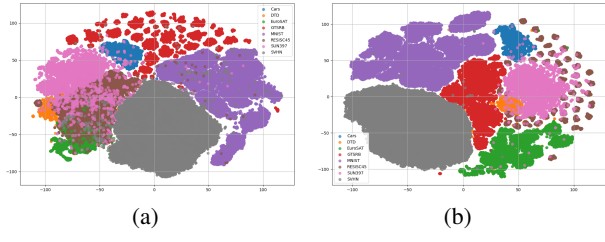

(a)                              (b)

Figure 1: Above figures show 2 dimensional visualization of embeddings of 8 tasks obtained from (a) GTSRB (b) RESISC45 fine-tuned weight.

## Conclusion and Future Works

In this work, we propose a training-free model merging framework that explicitly considers both the diversity and quality of source models. To the best of our knowledge, this is the first approach to incorporate the quality of source models—beyond merely the diversity of ensemble members—and to provide a generalization error bound for the merged solution. The framework is advantageous in its plug-and-play compatibility, enabling seamless application to existing methods.

We validate the effectiveness of our approach both theoretically and empirically. By deriving a generalization error bound and conducting experiments on image classification tasks, we demonstrate that merging diverse fine-tuned models (task vectors) of high quality leads to improved generalization ability. Furthermore, by strategically selecting models for averaging, our method achieves superior performance in generalization and robustness, even under non-i.i.d. settings where the assumption of identically distributed input data does not hold in the test domain.

Looking ahead, we aim to extend this framework to large language models, building upon the foundation laid by existing works in the domain of language foundation models. Additionally, there is potential to enhance diversity further by leveraging larger task pools, as suggested by (Wang et al. 2024). Future work will also seek to broaden the theoretical foundation by incorporating more intuitive measures for the generalization error bound, such as the flatness of the loss surface and the diversity of neural network weights, with connections to linear mode connectivity.

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
