# OpenReview forum: "Quality and Diversity Both Matters When Merging Models"
_AAAI.org/2025/Workshop/NeurMAD — AAAI 2025 Workshop NeurMAD Submission_

### Official Review · Reviewer_Db48 · 2024-12-27
**Clear and rigorous ideas, but with limited novelty**

**Rating:** 5
**Confidence:** 2

**Review:**

This paper addresses the significant challenges and limitations faced by current ensemble methods in achieving robust generalization under distribution shifts. It proposes a novel framework that incorporates selective models evaluated by error, supported by theoretical proof, and filtered by Determinantal Point Processes (DPP), taking into account both diversity and quality.

Strength:
 - The authors propose a novel kernel construction based on DPP to select models that balance both high performance (quality) and diversity.
- The authors first propose a generalization error bound and provide strong theoretical derivations to demonstrate that the previous weight-averaging methods for model aggregation can achieve a tighter generalization error bound by considering the contributions of both quality and diversity.
- The method is plug-and-play: This approach is designed to be easily applicable without requiring extensive modifications to existing systems.
- The method overcomes the limitations of traditional assumptions for i.i.d. data: It provides a solution that is not constrained by the i.i.d. assumption, making it more flexible and applicable to a wider range of scenarios.
- The output of experiments seems improved, even though the test dataset is small and there is a lack of benchmarks from other merging methods in the field for comparison

Weaknesses:
- Limited novelty: The proposed method mainly introduces the use of DPP for selecting ensemble models and provides theoretical support. However, the innovation is somewhat limited, as it primarily focuses on applying an existing technique (DPP) without introducing substantial new methodologies.
- Lack of effective comparison: The experiments lack comprehensive comparisons with other established methods in the field. Only the proposed method is added on top of the baseline, which makes it difficult to evaluate its performance relative to other merging techniques. The absence of benchmarks from other merging methods in the field undermines the robustness of the comparison.
- The derivation of the generalization error bound is also based on certain assumptions(like i.i.d), which may not hold in some cases, potentially affecting the practical applicability of the theoretical results.

---

### Decision · Program_Chairs · 2024-12-30

**Decision:**

Reject

**Comment:**

This work has a bit of novelty. We agree with the reviewer's comments.